# OpenReview forum: "Challenges with unsupervised LLM knowledge discovery"
_NeurIPS.cc/2024/Conference — Submitted to NeurIPS 2024_

### Official Review · Reviewer_DA2a · 2024-07-08

**Soundness:** 3
**Presentation:** 3
**Contribution:** 3
**Rating:** 6
**Confidence:** 2

**Summary:**

This paper reveals novel pathologies in existing unsupervised methods aimed at discovering latent knowledge from large language model (LLM) activations. Instead of extracting knowledge, these methods tend to identify the most prominent features of the activations.

The paper theoretically demonstrates that arbitrary features (not just knowledge) can satisfy the consistency structure of a popular unsupervised knowledge-elicitation method, namely contrast-consistent search. Additionally, the authors conducted a series of experiments showing that current unsupervised methods for discovering latent knowledge are insufficient. While the paper proposes potential future solutions, it does not provide a definitive solution to the problem with existing unsupervised methods.

**Strengths:**

Overall, the paper is well-written, and its theoretical and analytical contributions may be useful. I am impressed about the extensive experiments.

**Weaknesses:**

More experiments on other LLMs are needed to further validate the claim.

It would be better to offer possible solutions to address the problems in existing unsupervised methods.

**Questions:**

More experiments on other LLMs are needed to further validate the claim.

It would be better to offer some possible solutions to address the problems in existing unsupervised methods.

**Limitations:**

No solutions to address the problems in existing unsupervised methods.

---

> ### Author Rebuttal · Authors · 2024-08-05
>
> We thank the reviewer for their considerations and for highlighting how our work is useful and contains extensive experiments.
> Weaknesses:
> - We already considered three LLMs. We are somewhat limited by access to model internals to carry out this work (e.g. we can’t use APIs) and by licensing restrictions for some open-source models.
> - Don’t provide solutions to the issue: see top-level comment. In section 6 we provide desiderata that a solution should satisfy. It was beyond the scope of this work to provide a solution, rather to point out the problem through careful experiments and theory. Providing a solution would require another paper (and would be very difficult to do, as each desiderata requires solving a difficult open problem).

---

> > ### Comment · Reviewer_DA2a · 2024-08-09
> >
> > Although I gave an acceptable score, I am not an expert on the content of this paper. I gave a high score considering the well-written, so-called theoretical analysis, and comprehensive experiments. In my opinion, there may be some potential value in funding or observation: the latent knowledge discovered so far may be the most prominent feature of activation. On the other hand, I still think that this paper is an empirical analysis without possible solutions. In Section 6, the simple discussions may not solve the problem well because they do not have feasibility details. Besides, the opinions of other reviewers will affect my score.

---

### Official Review · Reviewer_cjB1 · 2024-07-09

**Soundness:** 3
**Presentation:** 4
**Contribution:** 3
**Rating:** 4
**Confidence:** 3

**Summary:**

This paper presents a careful study on existing methods for discovering the latent knowledge from large language models (LLMs), especially Contrastive-Consistent Search (CCS). The authors prove that CCS might not actually discover the knowledge of LLMs, instead, it could fit any features that satisfy certain conditions. Through a series of experiments, the authors further demonstrate that CCS could be distracted by random words, irrelavant texts like the character's opinion, and remain sensitive to the choice of prompt. Finally, the authors propose some general principles for the future works about unsupervised LLM task discovery.

**Strengths:**

Overall, the paper is well written and eazy to follow. The authors made interesting obervations about existing methods on knowledge discovery of LLMs. The theoretical analysis is well supported by the experiments. Sevaral guiding principles are also proposed for the future works. I think this paper would provide good information to the research community about unsupervised knowledge discovery of LLMs.

**Weaknesses:**

From my experience on unsupervised learning, I'd argue that the content of this paper *would not be sufficient to refute existing methods about unsupervised knowledge discovery (CCS)*. First of all, CCS is a method built on top of features from pretrained models. It'd definitely be sensitive to the features and thus also sensitive to the prompts, because features changes from different prompts (this could also be seen from the PCA visialization). Furthermore, as an unsupervised method, it'd be expected that the method might find multiple valid solutions, where only one of the solutions corresponds to the knowledge we are looking for. Taking the experiments from Section 4.2 as an example. The constructed dataset actually has two valid labels: the sentiment of the text and the sentiment of Alice. Depending on the optimization and the implicit bias of the algorithm, it could totally happen that an unsupervised method could found both valid labeling, or could only find one of them. I believe this is a common phenomenon shared by exsiting off-the-shelf unsupervised methods (like K-Means) cause they're searching for labels without supervision. From this perspective, I'd regard that this paper provides a method to construct "adversarial datasets" for CCS. However, it would not be a problem for CCS in practice.

Furthermore, the authors don’t provide solutions to this issue.

Also, I believe the mathematical notation in Section 3 could be simplified.

Minor issues: typo $c(x_i^+=1), c(x_i^+)=0$ in line 102

**Questions:**

Does the sensitivity of CCS come from the algorithm design, or come from the sensitivity of LLMs? As a baseline, if we add text "Alice think it's positive" at the end of every sentence, what would be the performance of (calibrated) zero-shot inference and in-context inference? If zero-shot and in-context inferences could also be easily distracted by "Alice think it's positive", then I don't think the problem discussed in this paper could be a clear shortcoming of CCS.

If the dataset exists multiple valid labelings (e.g., the texts from a group of people might reflect different gender, age or cultures), what do you think a proper unsupervised method should do on uncovering the latent knowledge?

What would be the performance of CCS on other langauge models (especially instruction-tuned models)? Would the model still be senstive to the prompts and could be easily distracted?

**Limitations:**

The authors have mentioned that this paper is focused on current methods and might not be directly applied to future works.

---

> ### Author Rebuttal · Authors · 2024-08-05
>
> We thank the reviewer for their comments, and for highlighting that our work provides useful information to the community about unsupervised knowledge discovery.
>
> Weaknesses:
> - Prompt sensitivity of CCS and other unsupervised methods (including PCA): this is a key point we were making in our paper, we don’t understand how it is a weakness. If we interpret the reviewer as claiming that this point is a priori obvious, then we would question why others using the method (see table 1) have not noticed this before or made attempts to account for it.
> - Multiple solutions: we agree that we have successfully shown multiple solutions, and moreover that we can say for sure that one of them (which we artificially introduce and can measure) is a meaningful undesired feature (rather than some arbitrary feature). That this happens in other unsupervised methods is exactly our point: CCS does no better than off-the-shelf unsupervised methods, which is a key counterclaim compared to the CCS paper’s claim of the effectiveness of their approach.
> - Adversarial datasets are not a problem for CCS in practice: we constructed adversarial datasets so that we could measure their effect. In the wild, there are likely other binary features of datasets that are naturally occurring, and highly salient, and CCS may well discover those as solutions instead. We believe the problem of multiple valid solutions will be a problem in practice.
> - Don’t provide solutions to the issue: see top-level comment. In section 6 we provide desiderata that a solution should satisfy. It was beyond the scope of this work to provide a solution, rather to point out the problem through careful experiments and theory. That would require another paper.
> - Typo: thanks we can fix it in revision.

---

> > ### Comment · Reviewer_cjB1 · 2024-08-09
> > **Acknowledgement**
> >
> > Thank you for the rebuttal. There're a few points that I want to mention.
> >
> > **Multiple solutions**: in Section 4.1-4.3, the authors manually created multiple valid solutions and show that CCS, like other unsupervised methods, could discover undesired features. But as I said, CCS is an **unsupervised method**. It's fine for an unsupervised method to discover any possible solutions, as long as the solution satisfies the pre-defined metric. *Unless further information is provided for steering the method, it's not possible to expect an unsupervised method to find the "exact desired knowledge" from multiple valid solutions*. Otherwise, how do you know "the sentiment of the sentence" is knowledge, while "Alice's opinion" is not knowledge? Thus I don't think the claim "we show that unsupervised methods detect prominent features that are not knowledge" is convincing.
> >
> > Besides, I'd like to see some experiments on real datasets that have multiple valid solutions. But the question I aksed is not replyed:
> > - If the dataset exists multiple valid labelings (e.g., the texts from a group of people might reflect different gender, age or cultures), what do you think a proper unsupervised method should do on uncovering the latent knowledge?
> >
> > **Sensitivety to the prompt**:  One of the major contribution of is paper is to show that CCS could be sensitive to the prompts. However, I'd expect deeper analysis about the sensitivity. Does it comes from the algorithm design, or comes from LLM itself? Thus, I asked two questions in the initial review (but also not got replied):
> > - Would (calibrated) zero-shot inference and in-context inference also be affected in this case?
> > - For other language models (especially instruction-tuned models), what would be the performance?
> >
> > **Don't provide solutions**: this would be a limit of the contribution of this paper. And the principle provided in Section 6 is not making perfect sense for unsurvised methods.
> >
> > Overall, I still have some concerns about this paper. And some questions I asked are not answered. Thus I will keep my score.

---

> > > ### Author Response · Authors · 2024-08-09
> > >
> > > **Multiple solutions**: our claim is more that CCS (and other unsupervised approaches) will not necessarily discover the intended feature which is underspecified by the pre-defined metric.
> > >
> > > **Real datasets**: it is hard to spot the discovery of unintended features in real datasets is because one does not know what feature to measure. That's why we did this in a controlled setup, where we know what to measure, with varying degrees of realism.
> > >
> > > **algo vs LLM**: this doesn't seem possible to answer cleanly as the methods operate on top of LLM features. We don't think 0/in-context inference bears much relevance to this issue. The promise of CCS is that it could be used eg as a lie detector and so it should work regardless of whether the LLM's response is correct/incorrect.
> > >
> > > **instruction-tuning**: this is similar to the above. An unsupervised knowledge discovery algorithm operating on model internals should not be that sensitive to finetuning details of the model.

---

### Official Review · Reviewer_1zMZ · 2024-07-12

**Soundness:** 3
**Presentation:** 3
**Contribution:** 2
**Rating:** 3
**Confidence:** 3

**Summary:**

This paper studies the failure modes of the method called "constraint-consistent search (CCS)" in knowledge discovery for language models. In particular, they showed: there is no unique identification on the minimizer of CCS, as there are a class of features achieves the optimal loss; demonstrated experimentally classic unsupervised methods detect features other than knowledge; discovered features are sensitive to prompt formats.

**Strengths:**

This paper points out a popular method's overlooked short-comings and presents both theoretical and experimental results to support that CCS may not be able to discover the true knowledge feature: 1. the observation on CCS loss is driven by xor operator rather than the feature is clever; 2. given the vast space of feasible features, CCS method is very sensitive to prompts and thus deserves more careful examination if to use CCS in practice.

**Weaknesses:**

The main weakness of the paper is its lack of novelty and potential impact to the field. The paper is more an analysis work on the application of a single method [1] proposed in 2023, which given the speed of ML innovation, it is hard to see long-term benefits of this criticism.  The general principles proposed in the discussion section (Section 6) are interesting and fit more into the line of proposing desiderata for the field - though in their current status, require more rigorous work.

[1] C. Burns, H. Ye, D. Klein, and J. Steinhardt. Discovering latent knowledge in language models without supervision. In The Eleventh International Conference on Learning Representations, 357 2023.

**Questions:**

Experiments:
The experiments lack variability. Section 4.1 - 4.3 are all experiments on modification of prompts to ensure increasing natural formalization of opinions in prompts. As the authors said in line 122 - 123, the learned probe depends on "prompt, optimization algorithm, or model choice".  Some experiments to show effects in optimization algorithm on knowledge discovery could be beneficial.

Typos:
- line 132 incomplete sentence: "Our experiments a structured didactically."
- line 46 first point in contribution - imprecise meaning on "arbitrary features satisfy the CCS loss equally well." - maybe more in the line of "arbitrary features achieve optimal CCS loss."

Above are some minor suggestions, though my main concern is on the potential impact of the paper to the field given it only challenges one recent method with lack of more-well rounded analysis on the general desiderata for the field.

**Limitations:**

Yes.

---

> ### Author Rebuttal · Authors · 2024-08-05
>
> We thank the reviewer for their analysis. The reviewer mentions in the strengths that the CCS method is popular, but suggests in the weaknesses that analysis showing shortcomings of the method is unlikely to have long-term benefit, which seems slightly contradictory.
>
> Weaknesses
> - Focus on CCS: see top-level comment. We consider other methods in our experiments, not just CCS. Reviewer cites innovation speed of ML broadly, but to our knowledge there has not been another unsupervised knowledge discovery method proposed since CCS’s publication. CCS has been published in top-tier ML conference, displaying the importance of our work highlighting its shortcomings. Table 1 in the appendix lists the 20 most-cited papers mentioning CCS and their usage. Our paper has implications for all of these works. These works are important for ML overall especially regarding honesty and interpretability.
> - We would be happy to develop further the discussion of section 6, highlight it as its own section and make the desiderata more detailed. This would be a minor addition in the revision.
>
> Questions
> - Experiments lack variability: our experiments varied the prompts, models and unsupervised methods. We could also vary optimization algorithms of the unsupervised methods but we think this is least likely to show an understandable effect because it is not well-understood which solution optimization algorithm will find, and whether that is just due to optimization failure/hyperparameter tuning, rather than that it will discover a meaningful undesired feature (what our experiments aim for). It was by design that experiments in 4.1-3 follow naturally from each other, in order to demonstrate the issue as simply as possible (though artificially), before making it gradually more realistic. We believe that a more random assortment of experiments would have shown similar results though make the paper less readable.
> - Typos: thanks, we will fix in the revision.

---

> > ### Comment · Reviewer_1zMZ · 2024-08-10
> >
> > Thank you for engaging with the rebuttal. I disagree with the opinion that an analysis on a well-cited paper in a popular field would automatically offers impactful contribution.
> > Unsupervised learning methods are difficult to provide identifiability guarantees without strong assumptions [1, 2], which the other methods tested in experiments, e.g. k-means, PCA are all such unsupervised learning methods, therefore it is not surprising they cannot discover true knowledge. As the paper with its current stand in proposing no concrete solutions, the potential contribution is to offer understanding about the field and next steps, e.g. is unsupervised learning method in general impossible to discover knowledge; based on the empirical findings, what would the paper suggest for the field to focus on. I think the desidera in Section 6 is somewhat on the line to do that, but was not well-explored in the main sections and should, in my opinion, be more well-studied. Therefore, I will keep the score.
> >
> > [1] Ilyes Khemakhem, Diederik Kingma, Ricardo Monti, and Aapo Hyvarinen. Variational autoencoders and nonlinear ica: A unifying framework. In International Conference on Artificial Intelligence and Statistics, pp. 2207–2217. PMLR, 2020a.
> > [2] Francesco Locatello, Stefan Bauer, Mario Lucic, Gunnar Raetsch, Sylvain Gelly, Bernhard Schölkopf, and Olivier Bachem. Challenging common assumptions in the unsupervised learning of disentangled representations. In International Conference on Machine Learning, pp. 4114–4124. PMLR, 2019.

---

### Author Rebuttal · Authors · 2024-08-05

A common theme in the reviews was that it challenges only a particular method, CCS. In fact, our experiments involved multiple other unsupervised methods which all suffer the same issue, which is a general issue, and a hard one to solve. Similarly common was that our paper should have aimed to solve the problem. We wanted the scope of our paper to just be to show this general problem is real, rather than to solve it. This would require a completely different kind of work and a separate paper. We were commended for listing desiderata that our analysis suggests are necessary for a solution, each of which we believe represents a significant hurdle, i.e. there will be no easy solution here.

---

### Decision · Program_Chairs · 2024-09-25

**Decision:**

Reject

**Comment:**

This paper identifies key failure modes of unsupervised knowledge discovery methods for language models, in particular "Constraint-consistent Search," including the unique identifiability issue and sensitivity to prompt formats. Based on the reviews and discussions, the main concern is that while these observations are important, they align with well-established limitations of unsupervised learning methods and language models. As such, the findings do not offer significant new insights or surprises to the field.

Additionally, although the paper highlights these issues, it does not propose any solutions or mitigation strategies, leaving the contribution primarily diagnostic rather than constructive. leading to limited potential impact.